# Oral Manifestations Associated with HIV/AIDS Patients

**DOI:** 10.3390/medicina58091214

**Published:** 2022-09-03

**Authors:** Sarah Monserrat Lomelí-Martínez, Luz Alicia González-Hernández, Antonio de Jesús Ruiz-Anaya, Manuel Arturo Lomelí-Martínez, Silvia Yolanda Martínez-Salazar, Ana Esther Mercado González, Jaime Federico Andrade-Villanueva, Juan José Varela-Hernández

**Affiliations:** 1Department of Medical and Life Sciences, Centro Universitario de la Ciénega, Universidad de Guadalajara, Ocotlán 47810, Mexico; 2Master of Public Health, Department of Wellbeing and Sustainable Development, Centro Universitario del Norte, Universidad de Guadalajara, Colotlán 46200, Mexico; 3Periodontics Program, Department of Integrated Dentistry Clinics, Centro Universitario de Ciencias de la Salud, Universidad de Guadalajara, Guadalajara 44340, Mexico; 4Prostodontics Program, Department of Integrated Dentistry Clinics, Centro Universitario de Ciencias de la Salud, Universidad de Guadalajara, Guadalajara 44340, Mexico; 5HIV and Immunodeficiencies Research Institute, Department of Medical Clinic, Centro Universitario de Ciencias de la Salud, Universidad de Guadalajara, Guadalajara 44340, Mexico; 6HIV Unit Department, Antiguo Hospital Civil de Guadalajara “Fray Antonio Alcalde”, Universidad de Guadalajara, Guadalajara 44280, Mexico; 7Department of Plastic and Reconstructive Surgery, Hospital Regional de la Zona No. 89, Instituto Mexicano del Seguro Social, Guadalajara 44190, Mexico; 8Pediatric Dentistry Program, Department of Integrated Dentistry Clinics, Centro Universitario de Ciencias de la Salud, Universidad de Guadalajara, Guadalajara 44340, Mexico

**Keywords:** oral manifestations, HIV, AIDS

## Abstract

Oral manifestations are early and important clinical indicators of Human Immunodeficiency Virus (HIV) infection since they can occur in up to 50% of HIV-infected patients and in up to 80% of patients at the AIDS stage (<200 CD4^+^ T lymphocytes). Oral health is related to physical and mental well-being because the presence of some lesions can compromise dental aesthetics, and alter speech, chewing, and swallowing, thus impacting the quality of life of patients. For this reason, it is necessary to integrate, as part of the medical treatment of HIV-positive patients, the prevention, diagnosis, and control of oral health. It is essential that health professionals have the power to identify, diagnose, and treat oral pathologies through clinical characteristics, etiological agents, and risk factors, both local and systemic. A diagnosis at an early stage of injury allows optimizing and prioritizing oral treatments, especially in acute pathologies, such as gingivitis and necrotizing periodontitis. In this group of patients, the development of strategies for the prevention, control, and reduction of these pathologies must be prioritized in order to reduce morbidity and mortality in this group of patients.

## 1. Introduction

Oral manifestations are the earliest indicators of HIV infection and can predict the progression of HIV/AIDS [1,2,3,4,5]. Oral lesions occur in up to 50% of HIV-infected patients and in up to 80% of AIDS patients [1]. Since oral lesions are considered the first clinical features of HIV infection, as well as highly predictive markers of immunosuppression, they can be useful for early testing, diagnosis, and treatment of HIV/AIDS patients [6]. The introduction of antiretroviral therapy (ART) allowed a significant decrease in the appearance of opportunistic infections, together with a decrease in mortality and an increase in the survival and quality of life in HIV-infected patients [7]. The decrease in some oral manifestations, such as oral candidiasis, Kaposi’s sarcoma, oral hairy leukoplakia, among others, are considered efficacy markers of ART [7]. However, ART decreases salivary flow and promotes oral dysbiosis, generating a modification in the microbiota and facilitating the colonization of atypical microorganisms [8,9]. These ART side effects favor the presence of oral conditions such as xerostomia, triggering more severe and refractory oral pathologies, such as periodontal disease and dental caries [10,11,12]. The presence of oral lesions associated with HIV/AIDS has a great impact on the quality of life of these patients because oral health is related to physical and mental health [1,2,3]. Some of the oral lesions are ulcerative and painful, which can cause a loss of taste and lead to the loss of dental organs, compromising dental aesthetics, altering speech, and making chewing and swallowing difficult, which further favors a state of malnutrition, emaciation, and increased alteration of the immune system. Due to this, early and suitable treatment of oral manifestations is required in order to reduce morbidity and mortality in this group of patients. Oral health is essential to prevent systemic complications, such as bacterial infections or septicemia, which can be fatal in immunocompromised people [1,2]. The objective of this article is to make a review of the most representative oral manifestations described in HIV/AIDS patients. All of the photos were acquired from patients at the HIV clinic of the tertiary care university hospital “Antiguo Hospital Civil de Guadalajara—Fray Antonio Alcalde” (a 1000-bed teaching hospital in Western Mexico). The appropriate recognition of the injuries by doctors and dentists could generate an earlier and timelier referral to the HIV/AIDS treatment service or vice versa.

## 2. Oral Candidiasis

Oral candidiasis is the most common oral lesion in HIV/AIDS-infected patients, with a wide prevalence range, starting from 17% up to 75% [3,7,13]. Its most frequent culprit agent is *Candida albicans*, although there are other related species, such as *C. glabrata*, *C. tropicalis*, *C. krusei*, *C guilliermondii*, *C. dubliniensis*, *C. lusitaniae*, *C. parapsilosis*, *C. pseudotropicalis*, *C. stellatoidea* [14]. The risk factors predisposing oral thrush development are the CD4^+^ T lymphocyte count <200 cells/μL, tobacco use, and wide spectrum antibiotic use or corticosteroids, which result in oral dysbiosis [4,5,13]. Additionally, other local conditions might favor developing this condition, such as (a) using partially removable dentures, which create a substantially acidic, moist, and anaerobic environment, increasing mucous membrane permeability, which favors the pathogen’s colonization capability; (b) decreases in the quality and quantity of saliva are conditions prone to *Candida* infection development; and (c) malnutrition, malabsorption, or a deficient diet, particularly hematinic deficiencies (such as iron, vitamin B12, and folic acid), which are factors that might predispose oral candidiasis via mucous membrane compromise [15]. Oral clinical manifestations related to *Candida* are variable and have been mainly identified in three clinical entities: pseudomembranous (oral thrush), erythematous candidiasis, and angular cheilitis [3].

### 2.1. Oral Thrush

Oral thrush lesions are composed of white “creamy” plaques, which are soft or gelatin-like patches that can be scraped off, unveiling an underlying erythematous, which is an ulcered, eroded, and commonly painful surface (Figure 1a–c) [4,15]. These patches are composed of hyphae, keratin waste, inflammatory cells, epithelial cells, bacteria, and fibrin [15]. Oral thrush lesions can develop anywhere along the mucous membrane. However, the most common sites associated with this condition are the mucous folding, the oropharyngeal cavity, and the lateral borders of the tongue’s dorsal surface [4].

### 2.2. Erythematous Candidiasis

Erythematous candidiasis lesions usually present as flat red lesions commonly located on the palate or the dorsum of the tongue. Plaques showing papillae or keratin loss might be found (Figure 2). These lesions are usually symptomatic, accompanied by an oral burning sensation or taste distortion, particularly associated with the consumption of salty or spicy meals or sour beverages [4,15,16].

### 2.3. Angular Cheilitis

Angular cheilitis usually appears as fissures or linear ulcerations of the commissures and is frequently associated with small white unilateral or bilateral patches, as well as the presence of intraoral edema [15,16]. Overall, 20% of cases of this disease are caused by *Candida* spp., while 60% of cases are related to mixed *C. albicans* and *Staphylococcus aureus* infection [15]. Some local predisposing factors associated with this candidiasis presentation are narrow vertical oral dimensions, as well as aging-associated highly-stretchable skin. These conditions can lead to salivary contamination on the skin, triggering the disease. Angular cheilitis can appear with or without oral candidiasis (Figure 3) [16].

Most forms of oral candidiasis are diagnosed through basic clinical characteristics. However, when a diagnosis is unclear, it can be confirmed through direct fungal culture or by a potassium hydroxide test. The goal of these diagnostic tools is to identify yeast or pseudohyphae. Generally, oral fungal cultures are reserved for patients who show no response to therapy or when antifungal resistance is suspected [16].

## 3. Periodontal Disease

Periodontal diseases shows up in HIV-infected patients with a variable prevalence of 27% up to 76% when the patient has AIDS [6,12]. Periodontal disease encompasses a disease spectrum triggered by the complex interactions of mixed polymicrobial infections taking place in a dental biofilm while facing the host’s immune response, resulting in inflammation of the gums and surrounding tissues. In severe stages of periodontal disease, the destruction of teeth support structures (gum tissue, cementum, periodontal fibers, and alveolar bone) can be observed [12,17]. The mechanisms underlying the destructive process imply direct damage owing to the dental bacterial biofilm, as well as indirect damage, secondary to the bacterially-triggered host’s immune response [16,17]. The pathogens contributing to these inflammatory conditions are *Porphyromonas gingivalis*, *Prevotella intermedia*, *Tannerella forsythia*, *Fusobacterium nucleatum*, *Campylobacter rectus*, etc. However, in recent years, HIV-infected patients have been recognized to develop a much more severe and refractory periodontal disease, which has been attributed to the involvement of multiresistant pathogens, such as *Pseudomonas aeruginosa*, *Acinetobacter baumannii*, *Escherichia coli*, *Klebsiella pneumoniae*, *Enterobacter faecalis*, *Clostridium clostridiiforme*, as well as diverse *Candida* spp. [9,18,19,20,21,22,23,24,25]. The most relevant HIV/AIDS-related periodontal diseases are linear gingival erythema, chronic periodontitis, necrotizing gingivitis, and necrotizing ulcerative periodontitis [16,26].

### 3.1. Linear Gingival Erythema

This condition is considered a form of gingivitis, which is clinically composed of a 2–3 mm erythematous streak along the gingival frame. It is commonly accompanied by diffuse red lesions or petechiae-like wounds scattered towards the apex of gums as well as the alveolar mucous membrane (Figure 4a,b). It usually appears along the anterior teeth, though being capable of growing towards the molars, altogether with mucosal bleeding and scarce dental biofilm [4,23,26]. Among the representative features of this injury are clinical insertion loss and the absence of ulcers or pain. Multiple *Candida* spp. have been associated with triggering this condition [16,23,26].

### 3.2. Periodontitis

Periodontitis is characterized by microbial association within a dental biofilm, as well as host-mediated inflammation, which triggers the destruction of tissue supporting the teeth [27]. Its clinical manifestation includes gum inflammation, bleeding-prone mucous membranes, periodontal pouch formation, loss of clinical insertion, plaque, as well as supragingival and subgingival stone formation. In more severe cases, even dental displacement might develop, as well as pathological dental migration (Figure 5a,b) [27,28]. In HIV-positive patients, the dental biofilm usually has a relevant microbial heterogeneity, involving unusual bacteria within the oral cavity, such as *P. aeruginosa*, *A. baumannii*, *E. coli*, among others [18,20], as well as diverse *Candida* spp., encompassing *C. albicans*, *C. glabrata*, *C. krusei*, *C. tropicalis*, and *C. dubliniensis*. The presence of these microorganisms has been correlated to ART and a CD4^+^ T lymphocyte count of <200 cells/μL [9]. Some risk factors triggering periodontitis include passive tobacco consumption, diabetes, CD4^+^ T lymphocyte count <200 cells/μL, and xerostomia, among others [16,27].

### 3.3. Necrotizing Gingivitis (NG)

This form of gingivitis is characterized by severe erythematous gingival tissue, covered by a necrotic white-colored pseudomembrane, which is composed of inflammatory cells, necrotic tissue, bacteria, fibrin, ulcers (with an interproximal crater aspect), bleeding-prone mucous membrane, foul-smelling breath, and pain [4,16,23,26,29]. Teeth located on the anterior sextant are the most common location of this disease. However, it can extend over to the posterior teeth [16,23,26,29]. Additional signs and symptoms might appear as well, such as malaise, lymphadenopathy, halitosis, and fever [30]. The predominant microorganisms of this condition are pleomorphic anaerobes, such as *Prevotella intermedia*, *Fusobacterium nucleatum*, plenty of spirochetes of the *Borrelia genus*, and gram-negative anaerobic bacteria as well. However, some clues lead to clinical findings representative of other species, including *T. medium*, *T. maltophilum*, *T. amylovorum*, *T. oralis*, *T. macrodentium*, *Synergistetes* cluster A, and *Jonquetella anthropic* [30].

### 3.4. Necrotizing Periodontitis (NP)

This ailment shares the same clinical manifestations as NG, together with acute and severe adjacent alveolar destruction, as well as spontaneous bleeding and deep, intense pain (Figure 6) [27]. The loss of the alveolar ridge is related to marginal necrosis, meaning that periodontal pouch formation is unlikely. On the other hand, exposed alveolar and interseptal bones are common [4,26]. Most NP cases are limited to a single or a number of teeth, although they can be generalized. Its aggressiveness is characteristic since it might present in up to a loss of 90% of insertion within 3 to 6 months. In HIV-positive patients, the dental biofilm is composed of microorganisms such as *Enterococcus* sp., *Enterobacter sakazakii*, *Enterobacter cloacae*, *Serratia liquefaciens*, *Klebsiella oxytoca*, and *C. albicans.* The identified risk factors for NUP are smoking, xerostomia, and a CD4^+^ T lymphocyte count of <200 cells/μL [23,26,31].

## 4. Xerostomia

Xerostomia is defined as the subjective feeling the patient senses within the oral cavity of poor salivation [32,33]. Its frequency is up to 39% in patients with a CD4^+^ T lymphocyte count of <200 cells/μL and 27% with a CD4^+^ T lymphocyte count of >200 cells/μL. Xerostomia is also associated with the consumption of some drugs, which include antidepressants, ART, anxiolytics, oral antidiabetic agents (mainly sulfonylureas), respiratory agents, quinine, antihypertensives (such as thiazides and calcium channel blockers), urinary anti-spasm agents, glucosamine, NSAIDs, opioids, ophthalmologic drugs, and magnesium hydroxide. Furthermore, another predisposing factor is head and neck radiotherapy [33]. This pathology is predominant in patients who have a CD4^+^ T lymphocyte count of <200 cells/μL and whom also have CD8^+^ T lymphocyte proliferation within the main salivary glands, resulting in its destruction [3,5,6,32].

Regarding physical examination, it might be identified as a saliva-scarce mouth floor, together with erythematous dry oral mucosal membranes and tongue, which might come along with fissures (Figure 7) [32]. This ailment might develop together with a swallowing and/or speaking impairment and a low spicy, sour, and crunchy flavor threshold. Additionally, patients complaining of taste distortion or trouble using dentures are common. Salivation impairment has been considered a risk factor for developing caries, as well as increasing the risk of oral infection, such as candidiasis and periodontal disease [33,34]. It might also favor mucositis, tongue fissures, dysgeusia, speech difficulties, halitosis, oral irritation, chewing and swallowing trouble, and even weight loss and cachexia [33,34,35].

## 5. Kaposi’s Sarcoma

Kaposi’s sarcoma is the most common oral HIV-associated neoplasia, appearing in up to 6% of patients, and has been notably decreasing due to the use of ART [36,37]. This endothelial angioproliferative neoplasia is caused by human herpesvirus 8, which is transmitted during anal intercourse or through blood and saliva [4,16,37].

Its pathological features might oscillate according to the location of the lesions (ganglia, mucous membranes, or skin) as well as its morphological stage (patch, plaque, and nodule), progressing from papules, which convalesce into red-purplish plaques, which might ulcer and cause nearby tissue destruction [38,39]. The early lesions are usually flat, red, and asymptomatic. However, together with its evolution, the lesion usually develops a darker color with convergent wounds (Figure 8a) [4,16,37]. In advanced stages, the lesion might appear as multiple firm purple nodules, which might disturb normal mouth function and cause symptoms owing to trauma or infection. The ulceration and local destruction of this entity lead to the extraction of some dental organs in some cases (Figure 8b). It predominantly appears on the hard palate, major salivary glands, and jawbone [39]. Having a viral HIV load greater than 5000 copies/mL is an associated risk factor. The presumptive diagnosis of this neoplasia is made according to clinical characteristics. However, a definite diagnosis requires histopathological analysis after performing a biopsy [4,16,37].

## 6. Recurrent Aphthous Ulcers

Recurrent aphthous ulcers are the most common oral non-traumatic wounds. Their frequency ranges from 5% (without ART) up to 10% (patients on ART). The presence of these lesions is associated with a CD4^+^ T lymphocyte count of <200 cells/μL [7,40]. Their cause is yet unidentified. However, some theories have linked these lesions to an immune complex vasculitis or even to ART (Zalcitabine and abacavir) [37,40]. This ailment can be classified into minor aphthous ulcers and major aphthous ulcers:

### 6.1. Minor Aphthous Ulcers

These kinds of ulcers tend to appear most frequently. They usually show up as a simple yet painful erythematous halo, coated with a yellow-grayish pseudomembrane, with a diameter ranging from 2 to 5 mm (Figure 9). Its most common location is on a non-keratinized oral mucous membrane, typically lingering from seven to ten days, healing later without any scarring [4,16,41].

### 6.2. Major Aphthous Ulcers

These kinds of ulcers are generally observed in severely immunosuppressed AIDS patients (T CD4+ ≤100 cells/μL). These injuries appear as craters with elevated borders, covered with a white-yellowish pseudomembrane, measuring between 1 and 5 cm in diameter (Figure 10) [16,41]. These ulcers generally appear on the lateral border of the tongue, soft palate, mouth floor, oral mucous membranes, and oropharynx (including both keratinized and non-keratinized mucosal surfaces). Ulcers of this kind are remarkably painful, particularly while consuming salty, spicy, sour, and hard or rough foods and beverages. They may last up to weeks, causing dysphagia, trouble speaking and chewing, developing an induration on their borders, and leaving retractile fibrous scars after healing [4,16,41].

## 7. Oral Hairy Leukoplakia

Oral hairy leukoplakia has a lower prevalence among HIV-positive patients without ART (10%) compared to patients on ART (18%); its appearance is common during the fourth decade of life (52.1%), especially in the male population (about 79.8%) [7,42]. This lesion is caused by Epstein-Barr virus infection, though it is also associated with fungal microbes, usually *Candida* spp. [43] It often appears as a well-limited white lesion of variable appearance, transitioning from a flat lesion to growing papillary processes, similar to hair. Unlike candidiasis, it cannot be scraped off from mucosal surfaces. It commonly appears on the lateral borders of the tongue (bi or unilateral). However, it might extend to the tongue’s dorsum and mouth floor. On some occasions, it can appear on the oropharyngeal mucosa [4,26,42,43,44] (Figure 11a,b). This entity is associated with a low CD4^+^ T lymphocyte count, tobacco use, as well as topical and systemic steroid use [4,42,43,44].

## 8. Oral Hyperpigmentation

Oral hyperpigmentation appears in up to 37% of patients who have a CD4^+^ T lymphocyte count of <100 cells/μL. The causes of this condition remain unknown, though it has been associated with ART, particularly zidovudine, a thymidine analog reverse transcriptase inhibitor. Other drugs, such as clofazimine and ketoconazole, have also been associated, among others [16,45]. Oral hyperpigmentation manifests as black or maroon papules, associated with intraleukocytic melanin or pigments within the basal cell membrane or lamina propria, with premature melanosomes. It can show up anywhere on oral mucosal membranes [16,46] (Figure 12a,b). According to reports, HIV-positive patients are more frequently related to illicit drug use compared to the general population (28% vs. 16%, respectively) [47]. Methamphetamine use has been linked to developing dental disease, which might be relevant. These patients can show up with precarious oral hygiene, xerostomia, hypochromic lip lesions, lateral sides of cheeks and palate, accompanied by rampant caries (so-called oral “meth sores”), aside from grinding-related excessive dental wasting (Figure 13) [47].

## 9. Oral Herpes Virus

Oral wounds associated with Human Herpes Simplex Type 1 Virus (HSV-1) appear commonly. They affect up to 20% of HIV-positive patients [3,4]. Recurrent intraoral outbreaks due to HSV-1 infection begin as multiple fragile grouped papules and vesicles with a diameter shorter than 3 mm. These lesions fragment themselves to generate tiny and painful ulcers. Most of the lesions appear on keratinized or partially keratinized mucosal membranes (lips, hard palate, and gums). These wounds are self-limited, disappearing within seven to thirty days, although leaving a scar; they are linked to a CD4^+^ T lymphocyte count <100 cells/μL [4,44,48].

## 10. Oral Warts

The prevalence of oral mucous membranes and skin warts caused by multiple subtypes of human papillomavirus (HPV) is about 4.6%. This value has been increasing within the ART era [7,46]. As a consequence, studies have suggested that drugs or combinations of ART drugs might be a risk factor for HPV oral infection development [46]. These wounds show up as papules or solid elevated nodules, with a cauliflower or wheat spike appearance, flat or with a pedicle [46] (Figure 14a,b).

## 11. Caries

Dental caries is considered the most common oral disease in HIV-positive patients, with a prevalence reported between 54% and 83% [3,49,50]. It is a multifactorial disease triggered by the interactions of dental biofilm, skin surface, sugary diets, and the host’s vulnerability [2]. Bacteria on the biofilm involved in caries onset and progression, mainly *Streptococcus mutans* and *Lactobacillus*, respectively, metabolize sugars, thereby producing acid substances which that decompose enamel and dentin [10].

Caries can affect one or multiple dental surfaces; however, studies on the clinical characteristics and behavioral patterns of dental caries among HIV patients are scarce. Rezaei-Soufi et al. showed in their study a significant difference in the number of carious surfaces, including roots and crowns, in HIV-positive patients compared to HIV-negative patients. However, the prevalence of root caries is not significantly different between the two groups [51]. Additionally, it has been suggested that the severity of dental caries increases significantly with age and the duration of ART [51]. Within recent years, it has been suggested that *C. albicans* might increase caries development in patients who have HIV/AIDS, considering its capability to produce lactic acid through carbohydrate fermentation and hydroxyapatite dental structure degeneration processes, which is complicated with greater severity and the progression of caries development [10,11].

On the other hand, saliva has an essential role in preventing dental caries development due to its antibacterial and antifungal properties. It also possesses pH-buffering features within the oral cavity through bicarbonate and phosphate. As well, it provides necessary calcium and phosphate substrates to maintain dental enamel integrity. Lastly, saliva is also capable of producing antibodies [34]. HIV infiltration, CD8^+^ T lymphocyte proliferation within salivary glands, as well as antiretrovirals decrease the quality and quantity of salivary flow and modifies the normal oral cavity microbiome [8]; for this reason, these are considered the main risk factors for dental caries development in HIV-positive patients. Adding to the already mentioned predisposing conditions, suboptimal oral hygiene, tobacco use, drugs, periodontal disease, and a carbohydrate-rich diet are relevant factors leading to substantial dental caries development in this population (Figure 15a,b) [34,50].

## 12. Conclusions

Due to the advent of antiretrovirals, lifespan prolongation has been achieved for people living with HIV. The presence of oral disease can have a relevant negative impact on the quality of life of these patients. As previously mentioned, oral health is strongly associated with physical and mental health; therefore, the integral care of these patients must include the observation, detection, and treatment of oral pathologies, which can be complex and diverse; and that, on many occasions, could represent a challenge for the clinician, due to the development of more than a single disease in individual patients. None of the lesions described is exclusive to HIV/AIDS patients; however, all of them present higher prevalence, severity, and progression in comparison to HIV-negative patients, especially in low CD4^+^ T lymphocyte counts and, in some cases, associated with the use of ART. Three groups of oral disease manifestation have been defined and linked to HIV/AIDS, according to their severity and clinical presentation: (a) Group 1: these oral lesions are strongly associated with HIV/AIDS and are composed of seven kinds of lesions: oral candidiasis, hairy leukoplakia, Kaposi’s sarcoma, linear gingival erythema, necrotizing ulcerative gingivitis, necrotizing ulcerative periodontitis, and non-Hodgkin’s lymphoma; (b) Group 2: this group includes atypical ulcers, salivary glands diseases, viral infection, such as cytomegalovirus, herpes simplex virus, papillomavirus, and herpes zoster virus; and (c) Group 3: this group includes lesions rarer than those on groups 1 and 2, such as squamous cell carcinoma and diffuse osteomyelitis [37]. Given the aforementioned statement, it is relevant to remove barriers within oral healthcare for people living with HIV/AIDS, considering optimal and integral care as a goal.

## Figures and Tables

**Figure 1 medicina-58-01214-f001:**
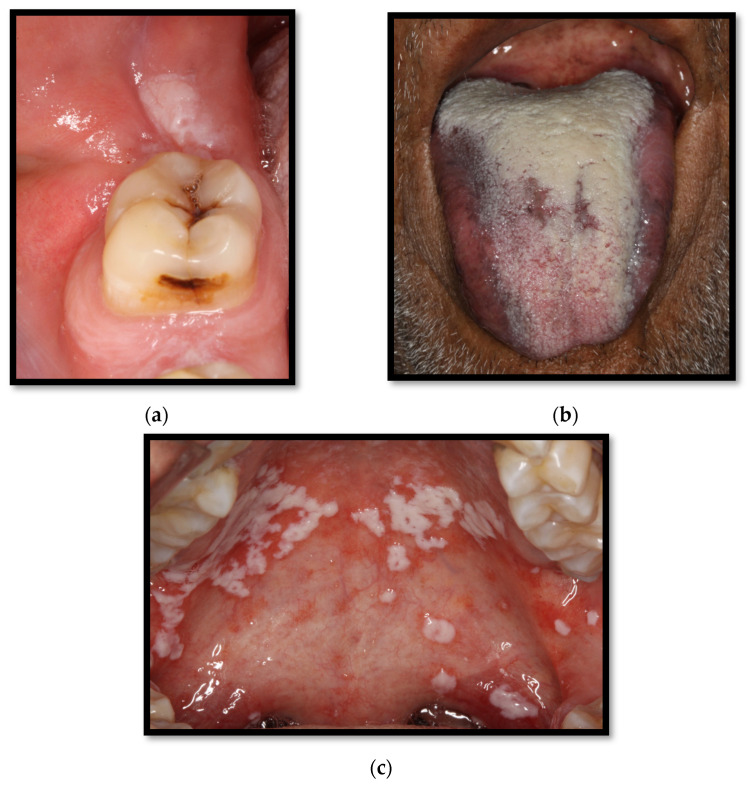
(**a**) Pseudomembranous candidiasis in the retromolar region; (**b**) pseudomembranous candida on the dorsum of the tongue; (**c**) pseudomembranous and erythematous candidiasis on the hard and soft palate.

**Figure 2 medicina-58-01214-f002:**
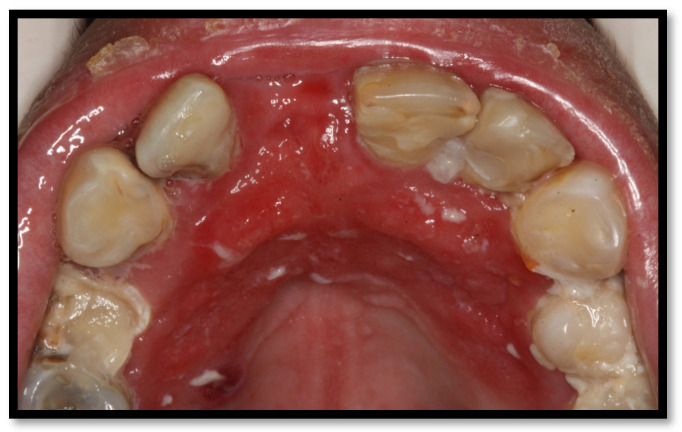
Erythematous candidiasis on the hard palate.

**Figure 3 medicina-58-01214-f003:**
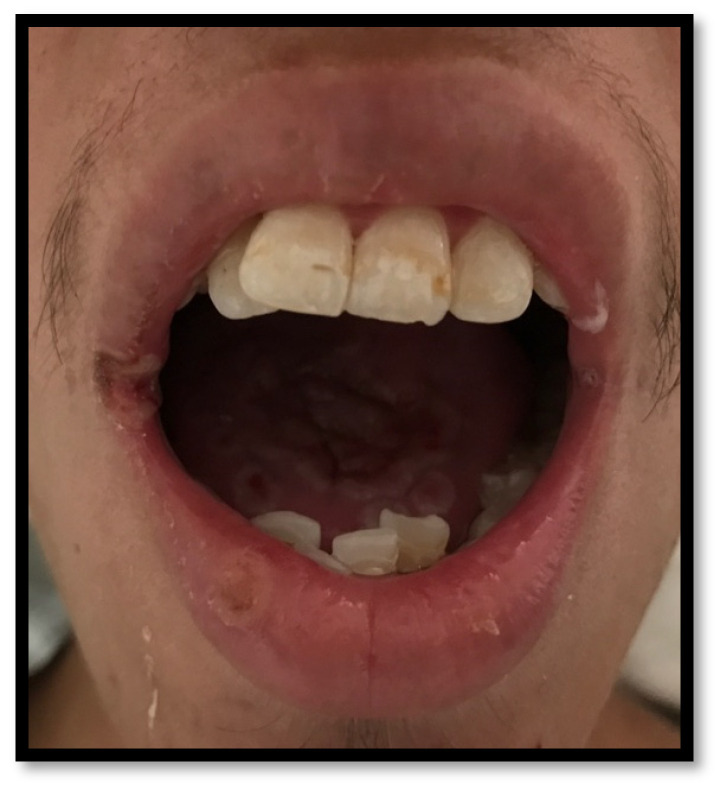
Angular cheilitis.

**Figure 4 medicina-58-01214-f004:**
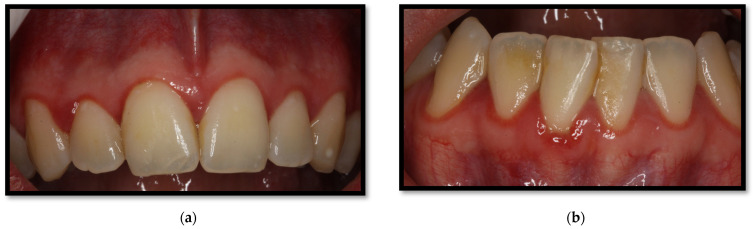
(**a**) Linear gingival erythema in the anterosuperior area (**b**) linear gingival erythema in the anteroinferior area.

**Figure 5 medicina-58-01214-f005:**
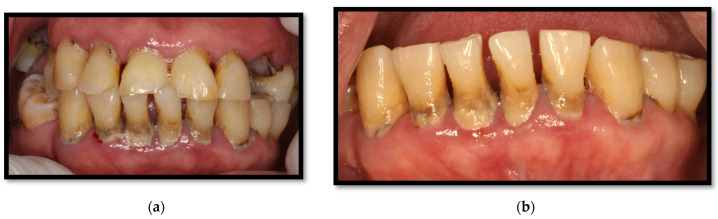
(**a**) Upper and lower arch chronic periodontitis (**b**) Anteroinferior sextant chronic periodontitis.

**Figure 6 medicina-58-01214-f006:**
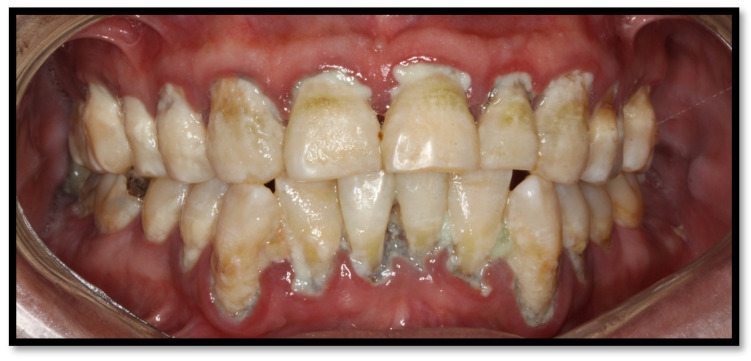
Anterior sextant necrotizing ulcerative periodontitis.

**Figure 7 medicina-58-01214-f007:**
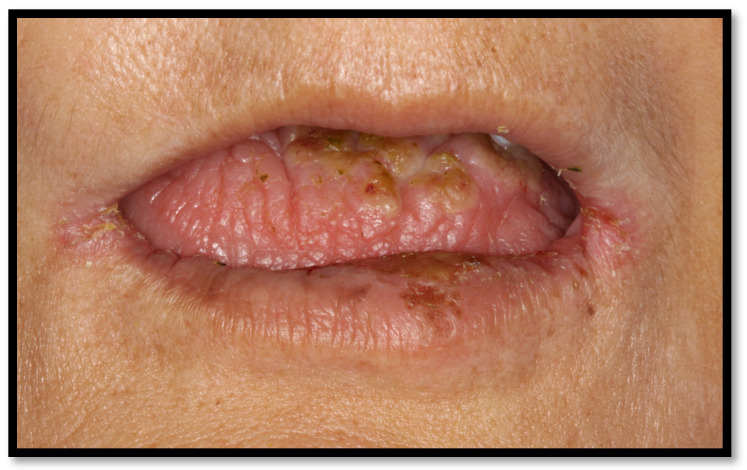
Tongue xerostomia and bilateral angular cheilitis.

**Figure 8 medicina-58-01214-f008:**
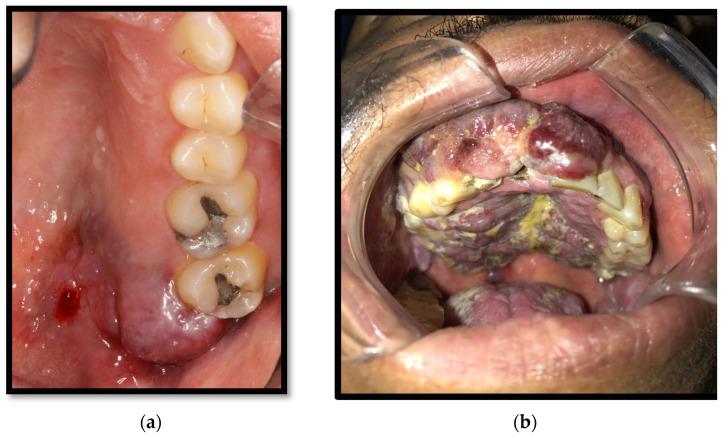
(**a**) Kaposi’s sarcoma in the soft palate and retromolar area; (**b**) Kaposi’s sarcoma of the gingiva, hard, and soft palate.

**Figure 9 medicina-58-01214-f009:**
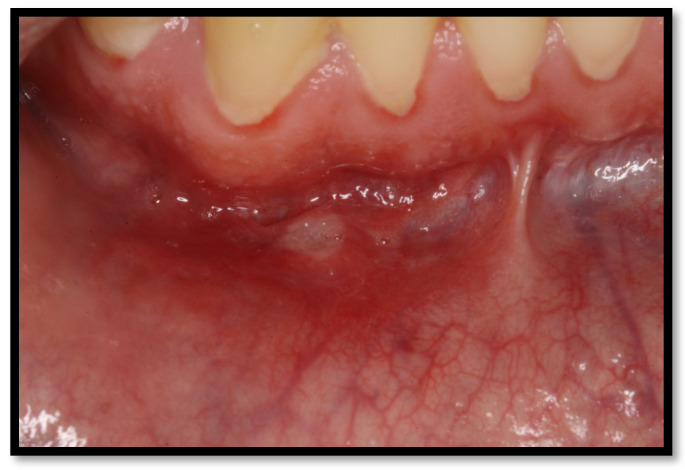
Minor aphthous ulcer in the lower vestibule mucosa.

**Figure 10 medicina-58-01214-f010:**
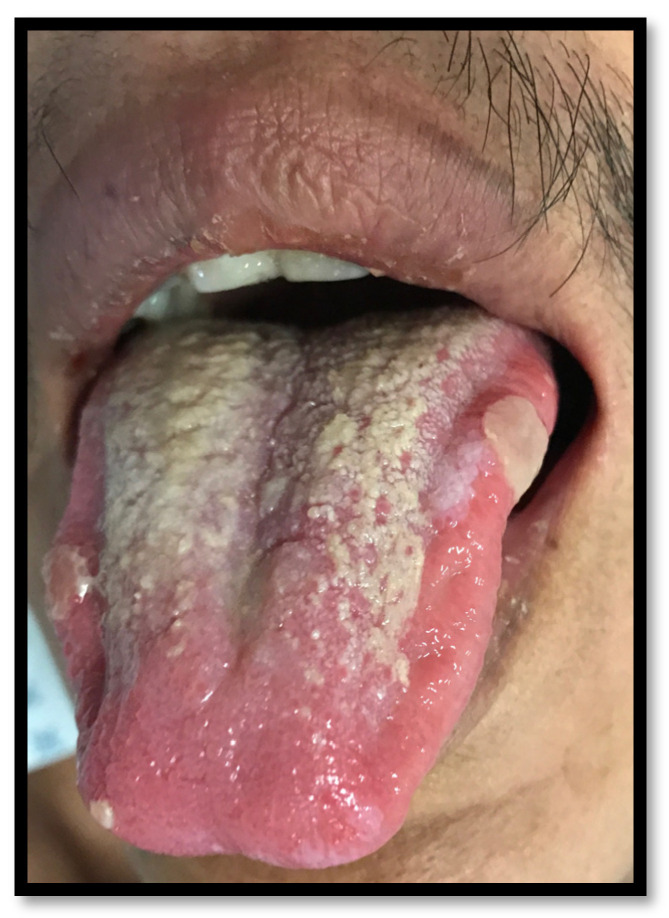
Major aphthous ulcers on the lateral borders of the tongue and coated tongue.

**Figure 11 medicina-58-01214-f011:**
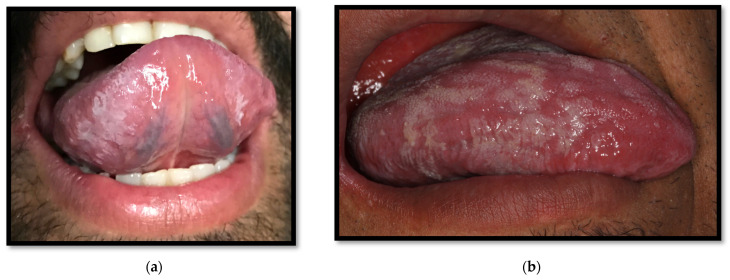
(**a**) Oral hairy leukoplakia on the underside of the tongue; (**b**) Oral hairy leukoplakia on the lateral border of the tongue.

**Figure 12 medicina-58-01214-f012:**
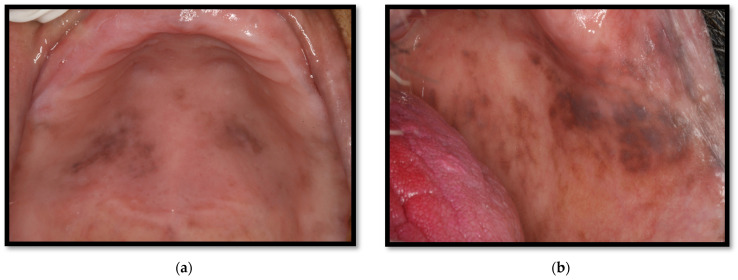
(**a**) Oral hyperpigmentation on the palate due to the use of the antiretroviral drug zidovudine; (**b**) Oral hyperpigmentation on the cheeks due to the use of the antiretroviral drug zidovudine.

**Figure 13 medicina-58-01214-f013:**
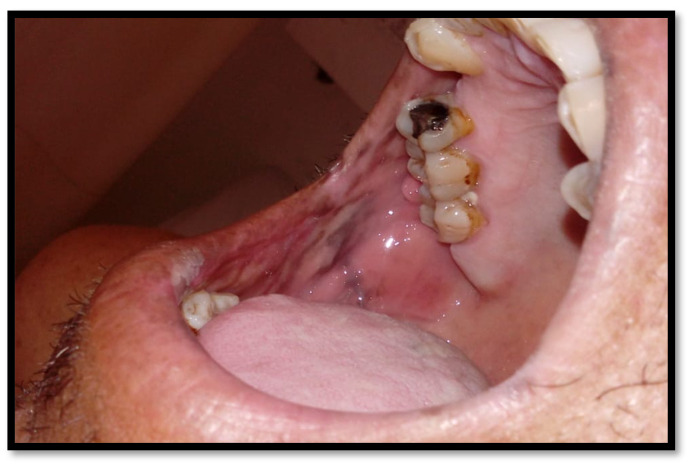
Hyperpigmentation in the lip mucosa, with severe dental destruction “methamphetamine mouth”.

**Figure 14 medicina-58-01214-f014:**
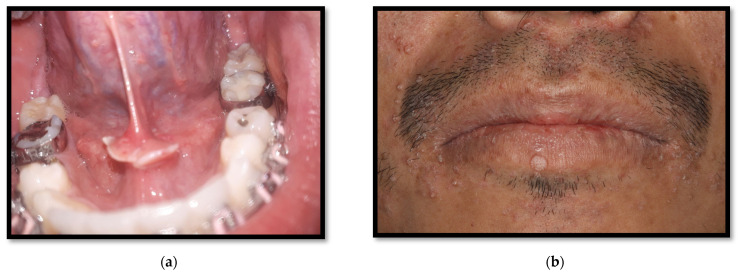
(**a**) Hyperplastic HPV lesion in the mucosa of the base of the lingual frenulum and sublingual caruncle; (**b**) wart due to HPV on the lower lip, with multiple umbilicated papules due to molluscum contagiosum and oily scaling lesions due to perioral seborrheic dermatitis.

**Figure 15 medicina-58-01214-f015:**
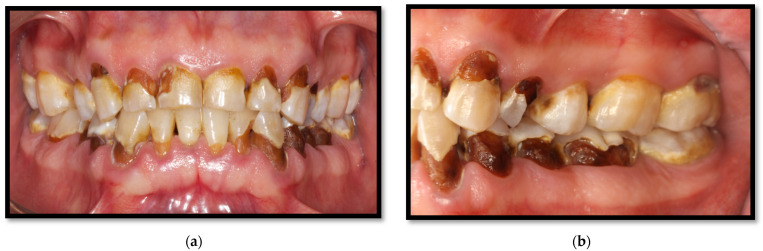
Multiple dental caries. (**a**) Front view, softened tissue in cervical and interproximal areas; (**b**) Lateral View, softened tissue in cervical areas.

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
