# Peer review of "Oral Manifestations Associated with HIV/AIDS Patients"

_medicina, 2022, doi:10.3390/medicina58091214_

Round 1

Reviewer 1 Report

The manuscript offers a comprehensive review of oral pathologies associated with anti-retroviral therapy (ART) for the treatment of HIV/AIDS.  The subject matter is extensive covering a wide range of oral pathologies in detail.  The authors provide a well referenced and illustrated overview of the field and update information that is highly relevant.  The study provides an important addition to the current literature.

Author Response

Point 1. The manuscript offers a comprehensive review of oral pathologies associated with anti-retroviral therapy (ART) for the treatment of HIV/AIDS.  The subject matter is extensive covering a wide range of oral pathologies in detail.  The authors provide a well referenced and illustrated overview of the field and update information that is highly relevant.  The study provides an important addition to the current literature.

I appreciate each of your comments and observations on the manuscript. Thank you very much.

Reviewer 2 Report

The article is easy to read and is a compilation of oral diseases prevalent in patients with HIV/AIDS patients. In this sense, it is a contribution for those who want to review these matters. I recommend the following -In general, in all the pathologies shown, but especially in Necrotizing gingivitis, Xerostomia, Minor and Major uphous ulcers, Oral hairy leukoplakia, oral herpes. Mention whether in HIV/AIDS patients the expression of the pathology is similar in patients without this underlying disease. or it is similar and then the emphasis is only on describing a pathology that the patient who presents it could have HIV/AIDS. -In Caries, the characteristics of these are different from a patient without this underlying pathology? -The figure shows 2 figures of multiple caries, but in the text where this disease is addressed, multiple caries is not deepened. -Figure 14a should be improved. Lastly, it is not clear if the photos included are of HIV/AIDS patients or if they are patients with the underlying pathology.

Author Response

The article is easy to read and is a compilation of oral diseases prevalent in patients with HIV/AIDS patients. In this sense, it is a contribution for those who want to review these matters. I recommend the following.

Point 1. In general, in all the pathologies shown, but especially in Necrotizing gingivitis, Xerostomia, Minor and Major uphous ulcers, Oral hairy leukoplakia, oral herpes. Mention whether in HIV/AIDS patients the expression of the pathology is similar in patients without this underlying disease. or it is similar and then the emphasis is only on describing a pathology that the patient who presents it could have HIV/AIDS.

The following was added to the manuscript:

“None of the lesions described is exclusive to HIV/AIDS patients, however, all of them present higher prevalence, severity and progression in comparison than in HIV-negative patients, especially in low CD4+ T lymphocyte counts and in some cases, associated with the participation of ART”

Point 2. In Caries, the characteristics of these are different from a patient without this underlying pathology?

Dental caries presents different characteristics in HIV-positive patients compared to negative ones.

The following was added to the manuscript:

“Caries can affect one or multiple dental surfaces, however, studies on the clinical characteristics and behavioral patterns of dental caries among HIV patients are scarce. Rezaei-Soufi et al. showed in their study a significant difference in the number of carious surfaces, including roots and crowns, in HIV-positive patients compared to HIV-negative patients. However, the prevalence of root caries is not significantly different between the two groups [52]. Additionally, it has been suggested that the severity of dental caries increases significantly with age and duration on ART [51]”.

Point 3. The figure shows 2 figures of multiple caries, but in the text where this disease is addressed, multiple caries is not deepened.

To our knowledge, there are no articles that delve into the subject of multiple caries. The articles available on the prevalence of dental caries in HIV patients has many limitations and does not show any pattern or specific behavior of this pathology (Rezaei et al., 2011; Kalanzi et al., 2019) .

The following was added to the manuscript:

“Caries can affect one or multiple dental surfaces, however, studies on the clinical characteristics and behavioral patterns of dental caries among HIV patients are scarce. Rezaei-Soufi et al. showed in their study a significant difference in the number of carious surfaces, including roots and crowns, in HIV-positive patients compared to HIV-negative patients. However, the prevalence of root caries is not significantly different between the two groups [52]. Additionally, it has been suggested that the severity of dental caries increases significantly with age and duration on ART [51].

Point 3. Figure 14a should be improved.

We tried to contact the patient who presented this pathology in the oral cavity, to improve the quality of the image. However, we did not locate the patient. Unfortunately, we do not have another photograph that represents this pathology inside the mouth.

Point 4. Lastly, it is not clear if the photos included are of HIV/AIDS patients or if they are patients with the underlying pathology.

All photos included in the manuscript are of HIV/AIDS patients recruited from the HIV clinic of the tertiary care university hospital “Antiguo Hospital Civil de Guadalajara— Fray Antonio Alcalde” (a 1,000-bed teaching hospital in Western Mexico).

The following was added to the manuscript:

“The objective of this article is to make a review of the most representative oral manifestations described in HIV/AIDS patients, all the photos where recruited from patients at the HIV clinic of the tertiary care university hospital “Antiguo Hospital Civil de Guadalajara— Fray Antonio Alcalde” (a 1,000- bed teaching hospital in Western Mexico). Appropriate recognition of the injuries by doctors and dentists could generate an earlier and timelier referral to the HIV/AIDS treatment service or vice versa.”

Round 2

Reviewer 2 Report

The observations made were answered satisfactorily and the changes in the manuscript are appropriate.